# Effects of L-Type Voltage-Gated Calcium Channel (LTCC) Inhibition on Hippocampal Neuronal Death after Pilocarpine-Induced Seizure

**DOI:** 10.3390/antiox13040389

**Published:** 2024-03-24

**Authors:** Chang-Jun Lee, Song-Hee Lee, Beom-Seok Kang, Min-Kyu Park, Hyun-Wook Yang, Seo-Young Woo, Se-Wan Park, Dong-Yeon Kim, Hyun-Ho Jeong, Won-Il Yang, A-Ra Kho, Bo-Young Choi, Hong-Ki Song, Hui-Chul Choi, Yeo-Jin Kim, Sang-Won Suh

**Affiliations:** 1Department of Physiology, Hallym University College of Medicine, Chuncheon 24252, Republic of Korea; doog0716@hallym.ac.kr (C.-J.L.); 2thqml@naver.com (S.-H.L.); bskang93@hallym.ac.kr (B.-S.K.); d22029@hallym.ac.kr (M.-K.P.); akqjqtj5@hallym.ac.kr (H.-W.Y.); m22091@hallym.ac.kr (S.-Y.W.); m2022087@hallym.ac.kr (S.-W.P.); m22525@hallym.ac.kr (D.-Y.K.); wjdgusgh1021@hallym.ac.kr (H.-H.J.); wonil4u@hallym.ac.kr (W.-I.Y.); 2Department of Physical Education, Hallym University, Chuncheon 24252, Republic of Korea; bychoi@hallym.ac.kr; 3Neuroregeneration and Stem Cell Programs, Institute for Cell Engineering, Johns Hopkins University School of Medicine, Baltimore, MD 21205, USA; akho3@jhu.edu; 4Department of Neurology, Johns Hopkins University School of Medicine, Baltimore, MD 21205, USA; 5Department of Neurology, Kangdong Sacred Heart Hospital, Seoul 05355, Republic of Korea; hksong0@hallym.oc.kr (H.-K.S.); yjhelena@hanmail.net (Y.-J.K.); 6Hallym Institute of Epilepsy Research, Chuncheon 24252, Republic of Korea; dohchi@hallym.ac.kr; 7Department of Neurology, Hallym University Chuncheon Sacred Heart Hospital, Chuncheon 24253, Republic of Korea

**Keywords:** seizure, amlodipine, L-type voltage-gated calcium channel, zinc, neuronal death, oxidative stress

## Abstract

Epilepsy, marked by abnormal and excessive brain neuronal activity, is linked to the activation of L-type voltage-gated calcium channels (LTCCs) in neuronal membranes. LTCCs facilitate the entry of calcium (Ca^2+^) and other metal ions, such as zinc (Zn^2+^) and magnesium (Mg^2+^), into the cytosol. This Ca^2+^ influx at the presynaptic terminal triggers the release of Zn^2+^ and glutamate to the postsynaptic terminal. Zn^2+^ is then transported to the postsynaptic neuron via LTCCs. The resulting Zn^2+^ accumulation in neurons significantly increases the expression of nicotinamide adenine dinucleotide phosphate (NADPH) oxidase subunits, contributing to reactive oxygen species (ROS) generation and neuronal death. Amlodipine (AML), typically used for hypertension and coronary artery disease, works by inhibiting LTCCs. We explored whether AML could mitigate Zn^2+^ translocation and accumulation in neurons, potentially offering protection against seizure-induced hippocampal neuronal death. We tested this by establishing a rat epilepsy model with pilocarpine and administering AML (10 mg/kg, orally, daily for 7 days) post-epilepsy onset. We assessed cognitive function through behavioral tests and conducted histological analyses for Zn^2+^ accumulation, oxidative stress, and neuronal death. Our findings show that AML’s LTCC inhibition decreased excessive Zn^2+^ accumulation, reactive oxygen species (ROS) production, and hippocampal neuronal death following seizures. These results suggest amlodipine’s potential as a therapeutic agent in seizure management and mitigating seizures’ detrimental effects.

## 1. Introduction

Epilepsy is a neurological disorder characterized by abnormal electrical activity in the brain, resulting in recurrent seizures. While the exact causes and mechanisms of epilepsy remain incompletely understood, significant progress has been made in understanding some of the contributing factors [1,2,3]. Seizures disrupt ion concentrations in the brain, including potassium and calcium, leading to the depolarization of neighboring neurons and the increased release of neuromodulators like zinc, which contribute to abnormal brain activity. Seizures can also damage various brain cells, such as astrocytes and microglia, disrupt microtubules, compromise the blood–brain barrier, and induce reactive oxidative stress. Zinc accumulation has also been observed in certain cases of epilepsy [4,5,6].

L-type voltage-gated calcium channels (LTCCs) play a critical role in regulating calcium influx in smooth muscle cells and neurons. The α1 subunit of LTCCs forms the channel pore and controls its opening. Upon membrane depolarization, the α1 subunit allows the entry of calcium ions (Ca^2+^) and other ions such as zinc (Zn^2+^) [7,8]. There are four isoforms of LTCCs, namely Cav1.1 (α1S), Cav1.2 (α1C), Cav1.3 (α1D), and Cav1.4 (α1F). In the brain, Cav1.2 and Cav1.3 are the predominant isoforms, with Cav1.2 being particularly abundant and playing a significant role in brain LTCCs. Cav1.2 has been associated with hippocampal long-term potentiation (LTP), a form of synaptic plasticity linked to learning and memory. It also participates in activity-dependent gene transcription [7,8,9,10,11,12,13].

Zinc is an essential mineral that plays crucial roles in various physiological functions in the body, including cell division, development, and DNA synthesis. Adequate zinc levels are necessary for optimal brain functioning and memory formation. However, disruptions in zinc homeostasis can have negative effects on brain function. Zinc deficiency can impair cognitive functions, particularly short-term memory. Conversely, excessive accumulation of zinc in brain cells can be detrimental. Following brain injuries such as seizures, ischemia, or trauma, an increase in neuronal death is observed. In these conditions, zinc accumulation within brain cells is believed to contribute to neuronal damage and cell death [14,15,16,17]. Zinc released during seizures can translocate to the intracellular space of postsynaptic neurons through various ion channels, including NMDA (*N*-methyl-d-aspartate) and AMPA (α-amino-3-hydroxy-5-methyl-4-isoxazolepropionic acid) receptors, as well as L-type voltage-gated calcium channels (LTCCs). This influx of zinc into postsynaptic neurons can increase intracellular Zn^2+^ concentration [18,19]. Elevated intracellular Zn^2+^ levels can interact with NADPH oxidase, an enzyme involved in the production of reactive oxygen species (ROS), such as superoxide radicals. This interaction can promote ROS production within neurons. ROS are highly reactive molecules that can cause oxidative stress and damage cellular components, including proteins, lipids, and DNA. Increased ROS production and oxidative stress resulting from zinc-induced interactions can have deleterious effects on neurons, ultimately leading to neuronal death. This process has been implicated in various neurodegenerative disorders, including Alzheimer’s disease, Parkinson’s disease, and ischemic brain injury [20,21].

Amlodipine is a dihydropyridine (DHP)-type drug commonly used as a calcium channel blocker (CCB). It specifically acts on L-type voltage-gated calcium channels (LTCCs) in various tissues, including smooth muscle cells in blood vessels and cardiac myocytes [22,23]. Amlodipine is primarily indicated for hypertension and angina. By reducing peripheral vascular resistance and improving coronary blood flow, amlodipine lowers blood pressure and relieves angina symptoms. Recent research has explored the potential use of amlodipine beyond vascular diseases and into the field of brain diseases [24,25].

During seizures, there is excessive activation of neurons and other cells in the brain, leading to an overload of calcium ions (Ca^2+^) and zinc ions (Zn^2+^). This overload can contribute to cell death and neuronal damage. The excessive influx of Ca^2+^ and Zn^2+^ into brain cells is facilitated by calcium channels, including L-type voltage-gated calcium channels (LTCCs). Amlodipine, as a dihydropyridine (DHP)-type calcium channel blocker, specifically binds to and blocks the Cav1.2 α1 subunit, which is the main component of LTCCs. By inhibiting the activity of Cav1.2 channels, amlodipine can effectively reduce the influx of calcium ions into brain cells. Additionally, by blocking LTCCs, amlodipine may indirectly affect the accumulation of Zn^2+^ in brain cells. As previously mentioned, excessive activation of neurons and cells during seizures can lead to the release of zinc ions, and the entry of Zn^2+^ into cells is facilitated by calcium channels, including LTCCs. By inhibiting LTCCs, amlodipine may help reduce the influx of Zn^2+^ into brain cells, potentially mitigating the detrimental effects of zinc overload [26,27,28].

Some reports have suggested the neuroprotective effects of amlodipine in central nervous system diseases and its potential use as an anticonvulsant in acute seizures [29,30,31,32,33]. Based on these findings, we hypothesize that treatment with amlodipine after pilocarpine-induced seizures in an animal model may lead to a reduction in oxidative stress, zinc accumulation, astrocyte and microglia over-activation, blood–brain barrier damage, and microtubule damage, and to an increase in cognitive function.

We hypothesize that amlodipine may exert a neuroprotective effect in seizure conditions. To test this hypothesis, we employed a controlled experimental design involving four groups: a seizure-vehicle group, a seizure-amlodipine group, and two sham controls. Our method included administering amlodipine post-seizure induction, followed by quantitative assessments using NeuN (Neuronal Nuclear) and DAPI (4′,6-diamidino-2-phenylindole) staining, to evaluate neuronal survival and integrity. Our findings suggest that amlodipine treatment leads to an increase in NeuN-positive cells compared to the seizure-vehicle group, indicating a potential protective effect against seizure-induced neuronal damage. However, these numbers did not reach the levels observed in the sham groups, suggesting a partial mitigation effect.

## 2. Materials and Methods

### 2.1. Ethics Statement and Care of Experimental Animals

The present study adhered to ethical guidelines for animal research and was approved by the Animal Research Committee at the College of Medicine at Hallym University (protocol number: Hallym 2021-39). Adult male Sprague Dawley rats (SD-Rats) were obtained from DBL Co. in Chungcheongbuk-do, Republic of Korea. The rats used in the experiment were approximately 8 weeks old and had an initial weight of 300–350 g. Strict control was maintained over environmental conditions, including a temperature of 20 ± 2 °C, humidity levels of 55% ± 5%, and a 12 h light/dark cycle.

### 2.2. Seizure Induction

Seizures were induced in rats through the administration of pilocarpine (25 mg/kg, i.p.) [34,35,36,37]. Prior to the injection of pilocarpine, lithium chloride (LiCl, 127 mg/kg, i.p.) was intraperitoneally administered, to enhance the action of muscarinic receptors. Additionally, scopolamine (2 mg/kg, i.p.) was injected 30 min before pilocarpine to reduce saliva production. The severity of status epilepticus (SE) was assessed using the Racine stage method, which categorizes seizures into five stages based on the following observed behaviors: 1. facial movement, 2. head nodding, 3. forelimb clonus, 4. rearing, and 5. falling. Following the occurrence of forelimb clonus and rearing, our pilocarpine-induced seizure model consistently reached a seizure severity of over phase 4, characterized by rearing, as per the Racine scoring system. This confirms the successful induction of seizures in our experimental setup [38,39]. Diazepam (10 mg/kg, i.p.), a commonly used antiepileptic medication for terminating or suppressing seizures, was administered one hour later. This experimental design allowed for the controlled induction and assessment of the severity of status epilepticus in the rats, followed by the administration of diazepam to manage and control the seizures.

### 2.3. Amlodipine Administration

The animals were divided into four groups: sham-vehicle, sham-amlodipine, seizure-vehicle, and seizure-amlodipine. In the amlodipine groups, amlodipine was administered orally at a dose of 10 mg/kg, dissolved in 0.9% saline. The first dose of amlodipine or saline was administered 1 h after seizure induction, and the treatment was continued daily for 7 days. For the behavior test to check cognitive function, the seizure groups received amlodipine or saline 1 h after seizure induction, and the administration was performed once a day for 12 days. For the TSQ (N-(6-methoxy-8-quinolyl)-para-toluene sulfonamide) staining procedure to check zinc accumulation, the seizure-amlodipine group received amlodipine (10 mg/kg, p.o., Daewoong, Dr.Reddy's lab, Korea) 1 h after seizure induction, while the seizure-vehicle group received an equivalent volume of 0.9% saline at the same time [40].

### 2.4. Brain Sample Preparation

The rats that experienced seizures were euthanized at two time points, 24 h and 7 days after seizure induction. Anesthesia was induced by administering urethane (1.5 g/kg, i.p.). To preserve the brain tissue, the animals were perfused with 0.9% saline, followed by 4% paraformaldehyde. The brains were carefully extracted and post-fixed in 4% paraformaldehyde for 1 h. Following fixation, the brains were transferred to a 30% sucrose solution and allowed to sink until they reached the desired consistency. After two days, the brains were sectioned into 30μm thick slices using a cryostat microtome (CM1850; Leica, Wetzlar, Germany).

### 2.5. Detection of Zinc Accumulation

After a 24 h period following seizure induction, the rats were anesthetized with urethane (1.5 g/kg, i.p.), and the brains were harvested without perfusion. To preserve the brain tissue, the brains were rapidly frozen in dry ice for approximately 1 min and then stored at −80 °C. Using a cryostat, brain samples were cut into 10 μm thick slices. These sliced sections were immediately mounted onto pre-coated slides (Fisher Scientific, Pittsburgh, PA, USA) and allowed to dry at room temperature for 1 h. Subsequently, the samples were stained with a 0.001% solution of TSQ (N-(6-methoxy-8-quinolyl)-para-toluene sulfonamide) obtained from Molecular Probes, Eugene, OR, USA [40]. The staining process lasted for 1 min. After staining, the samples were washed with a 0.9% saline solution for 1 min. The stained samples were observed using a fluorescence microscope (Olympus, Tokyo, Japan) equipped with UV light under a 360 nm wavelength and a 500 nm long-pass filter. This allowed for the visualization of the TSQ-positive cells, indicating zinc accumulation. To quantify the TSQ-positive cells, blind quantification was performed. The cells were counted without prior knowledge of the experimental conditions, ensuring an unbiased assessment of the results.

### 2.6. Detection of Oxidative Stress in the Hippocampal Region

For the 4-hydroxyl-2-nonenal (4HNE) immunohistochemistry assay, brain tissue samples were washed three times in 0.01 M phosphate-buffered saline (PBS) for 10 min each. A pretreatment step was performed to eliminate any blood present in the brain tissue’s blood vessels. The tissue samples were sequentially treated with distilled water, 90% methanol, and 30% hydrogen peroxide for 15 min each. After each treatment, the samples were washed three times for 10 min each in 0.1 M PBS. The brain tissue samples were then incubated overnight at 4 °C with a 4HNE-specific primary antibody solution. The primary antibody used was mouse anti-4HNE serum, obtained from Alpha Diagnostic Intl. Inc., San Antonio, TX, USA, diluted at a ratio of 1:500 in PBS containing 0.3% Triton X-100. Following the overnight incubation, the brain tissue samples were washed three times for 10 min each in 0.01 M PBS. Next, the samples were incubated with a secondary antibody solution containing donkey anti-mouse IgG conjugated with Alexa-Fluor-594 at a dilution of 1:250, obtained from Invitrogen, Grand Island, NY, USA. This incubation step was carried out at room temperature for 2 h. After incubation with the secondary antibody, the brain tissue samples were washed three times for 10 min each in 0.01 M PBS. Finally, the tissue samples were mounted onto slides, cover-slipped (Fisher Scientific, Pittsburgh, PA, USA) using DPX (Sigma-Aldrich Co., St. Louis, MO, USA) mounting medium obtained from Sigma-Aldrich Co., St. Louis, MO, USA, and observed using a fluorescence microscope from Olympus, Tokyo, Japan. To quantify the fluorescence intensity of the 4HNE staining, we used the ImageJ software (version 1.47c; NIH, Bethesda, MD, USA), developed by the National Institutes of Health, Bethesda, MD, USA, to quantify the fluorescence intensity of the 4HNE staining. Specifically, we analyzed the intensity within defined regions of interest in the tissue sections. The software allowed us to calculate the mean fluorescence intensity for these regions, and we then averaged these values to represent the overall fluorescence intensity for each experimental group.

### 2.7. Immunofluorescence Assay

To evaluate the impact of amlodipine on neuronal cells, an immunofluorescence assay was conducted. Brain tissue samples underwent a series of procedures, outlined as follows: First, the samples were washed in 0.01 M PBS for 10 min, and this process was repeated three times. Subsequently, a pretreatment solution, composed of distilled water, 90% methanol, and 30% hydrogen peroxide, was applied to the brain tissues for 15 min. After the pretreatment, the tissues were washed in 0.01 M PBS for 10 min, and this step was repeated three times. The brain tissue samples were then incubated overnight at 4 °C with primary antibodies. The specific primary antibodies used, along with their respective dilutions, in this study were as follows: rabbit anti-MAP2 (1:200, Abcam), rabbit anti-GFAP (1:1000, Abcam, Cambridge, UK), goat anti-C3 (1:300, Invitrogen, Boston, MA, USA), goat anti-Iba1 (1:500, Abcam, Cambridge, UK), mouse anti-CD68 (1:100, Bio-rad, California, USA), mouse anti-NeuN (1:500, Millipore, Billerica, MA, USA), and rabbit anti-Cav1.2 (1:300, Alomone labs, Jerusalem BioPark, Jerusalem). All primary antibodies were diluted in PBS containing 0.3% Triton X-100 (PBS-T). Following the overnight incubation, the brain samples were washed three times for 10 min each in 0.01 M PBS. Subsequently, the brain tissue samples were subjected to a 2 h incubation with a secondary antibody solution in PBS-T. The secondary antibodies used were specifically selected to align with the host species of the primary antibodies and were conjugated with fluorescent markers. However, the precise details of the secondary antibodies were not provided. After incubation with the secondary antibodies, the brain tissue samples underwent three washes of 10 min each in 0.01 M PBS. The washed brain samples were then mounted onto slides and cover-slipped (Fisher Scientific, Pittsburgh, PA, USA) using DPX (Sigma-Aldrich Co., St. Louis, MO, USA) mounting medium and observed using a fluorescence microscope from Olympus, Tokyo, Japan. For the fluorescence intensity measurement of brain tissue samples, we utilized ImageJ software (version 1.47c; NIH, Bethesda, MD, USA). The average value of the mean fluorescence intensity was determined and reported. Overall, this immunofluorescence assay provided insights into the effects of amlodipine on various neuronal markers, including MAP2, GFAP, C3, Iba1, CD68, NeuN, and Cav1.2, in the brain tissue samples.

### 2.8. Immunohistochemistry Assay

To evaluate live neuron detection and blood–brain barrier (BBB) disruption in brain tissue, the following steps were conducted, according to the immunofluorescence assay protocol. First, brain tissue samples were washed and pretreated as described. Then, the samples were incubated overnight at 4 °C with a primary antibody solution, containing mouse anti-NeuN (1:500, Millipore, Billerica, MA, USA) and 0.3% Triton-X in PBS. After the overnight incubation, the brain samples were washed three times with 0.01 M PBS. Subsequently, the samples were incubated with a secondary antibody solution containing anti-mouse IgG (1:250, Vector, Burlingame, CA, USA) and 0.3% Triton-X in PBS for 2 h at room temperature. This step allowed for the detection of IgG leakage following a seizure. For the analysis of IgG leakage, brain samples underwent the same pretreatment and washing steps as mentioned above. Then, the samples were incubated with a secondary antibody solution containing anti-rat IgG (1:250, Vector Labororoid, Burlingame, CA, USA) and 0.3% Triton-X in PBS for 2 h at room temperature. After the secondary antibody incubation, an ABC (avidin–biotin complex, Vector, Burlingame, CA, USA) solution was applied to the brain samples at room temperature for 2 h, helping to amplify the signal from the primary antibody. The brain samples were then colored with a 3,3′-diaminobenzidine (DAB, Sigma-Aldrich Co., St. Louis, MO, USA) solution in 0.01 M PBS buffer for 2 min. This reaction produced a brown color, as the DAB reacted with the ABC complex. The colored brain samples were mounted onto slides (Fisher Scientific, Pittsburgh, PA, USA) using a Canada balsam mounting medium (Junsei Chemical, Chuo-ku, Tokyo, Japan), and they were cover-slipped. To quantify the number of NeuN-positive cells and assess IgG leakage, we have now included a detailed description of the image analysis technique used for quantifying staining intensity, which serves as an indicator of blood leakage. This includes the ImageJ software used, the parameters set for analysis, and the approach for selecting regions of interest. The image analysis software program ImageJ (version 1.47c; NIH, Bethesda, MD, USA) was employed. This software facilitated the measurement of staining intensity and the quantification of the desired markers. By following this protocol, the immunohistochemistry assay enabled the detection of live neurons in the area of the hippocampus (CA1, CA3, hilus, subiculum) using the NeuN marker and the assessment of BBB disruption by analyzing IgG leakage in all hippocampal regions.

### 2.9. Behavior Test

#### 2.9.1. Barnes Maze Test

To evaluate the recovery of spatial cognitive ability after seizures, we conducted a study where rats were treated with amlodipine and subjected to the Barnes maze test. The rats, which had been epileptic for a week, were divided into two groups: one group received amlodipine treatment from the first day of seizure induction, while the other group had a rest period before starting the treatment. The Barnes maze test involved a circular board with multiple holes, with only one hole open and a cage placed underneath it. The rats were assessed over several trials, with each trial consisting of the rat being placed in a black cylinder, disoriented, and timed as they found the opening hole. After five days of testing, the rats were euthanized, and samples were stained using the previously mentioned method.

#### 2.9.2. Adhesive Removal Test

We examined cognitive recovery post-seizure using amlodipine in rats, employing both the adhesive removal test and the Barnes maze test on the same days. In the adhesive removal test, rats had 1 cm tape pieces on their paws; if removal exceeded 2 min, we intervened. This test assessed fine motor skills and sensory perception. Multiple trials were conducted for each rat. The Barnes maze test evaluated spatial learning and memory. The combination of these tests allowed for a comprehensive assessment of cognitive function recovery, shedding light on the effects of amlodipine treatment.

### 2.10. Data Analysis

The data in this study are reported as the mean ± SEM. To compare the vehicle-treated and amlodipine-treated groups, statistical analyses were conducted using the Mann–Whitney U test or the Kruskal–Wallis with post hoc Bonferroni. Behavioral data were assessed for variance using an ANOVA. All data were analyzed using IBM, from the Statistical Package for the Social Sciences (IBM SPSS statistics version 25, Chicago, IL, USA) software. Moreover, a blind test approach was implemented during the statistical analysis to reduce bias.

## 3. Results

### 3.1. Amlodipine Administration Reduced Cav 1.2 Activation and Zinc Accumulation in Neurons Following Seizure

In this study, we quantified the expression of the Cav 1.2 calcium channel as an indirect indicator of altered calcium influx in neuronal cells [41,42]. Our objective was to examine the effects of amlodipine, an inhibitor of L-type voltage-gated calcium channels (LTCCs), on Cav1.2 channel expression and zinc accumulation in the hippocampal region of the brain. Cav1.2 channels play a crucial role in the influx of various ions, including Ca^2+^ and Zn^2+^, and their activation during seizures leads to Zn^2+^ accumulation in neuronal cells.

To evaluate Cav1.2 channel expression, we performed immunofluorescence staining of brain samples using an assay that specifically targeted Cav1.2 channels in NeuN-positive cells, representing live neurons. We compared the expression levels between the group treated with amlodipine after status epilepticus (SE) and the SE group treated with a vehicle, as well as the vehicle-only groups. Our findings revealed that Cav1.2 expression was significantly higher in the CA1 and CA3 regions of the hippocampus in the vehicle-treated SE group, compared to the amlodipine-treated SE group and the vehicle-only groups. Interestingly, the expression levels of Cav1.2 channels in the SE amlodipine-treated group were comparable to those in the sham-treated group. Thus, our results demonstrate that amlodipine effectively reduced Cav1.2 channel expression during seizures (Figure 1A–D).

Furthermore, we conducted TSQ staining to assess zinc accumulation in the hippocampal region following pilocarpine-induced seizures. Seizures result in excessive neuronal activation, leading to neuronal damage and the subsequent accumulation of zinc within neurons. By blocking LTCCs, which are ion channels in neurons, our aim was to mitigate zinc accumulation and its potential detrimental effects on neuronal survival. Neither the sham-vehicle nor the sham-amlodipine groups showed significant zinc accumulation in neurons. However, within the seizure groups, a notable zinc accumulation was observed. Crucially, the seizure-vehicle group exhibited a twofold increase in the number of neurons with positive zinc accumulation compared to the seizure-amlodipine group. These findings highlight the efficacy of amlodipine in reducing Cav1.2 channel expression and diminishing zinc accumulation in neurons, potentially contributing to neuroprotection against cell death (Figure 1E,F).

In summary, our study demonstrates that amlodipine treatment during seizures effectively downregulates Cav1.2 channel expression and limits zinc accumulation in neurons. These findings underscore the potential neuroprotective properties of amlodipine, highlighting its therapeutic value in mitigating neuronal damage associated with seizures.

### 3.2. Administration of Amlodipine Decreased Reactive Oxidative Stress (ROS) after Seizure

To investigate the impact of amlodipine on reactive oxidative stress (ROS) in neuronal damage, we examined the expression of ROS in the hippocampal region one week after pilocarpine-induced seizures. ROS, triggered by factors such as neuronal over-activation and ion accumulation during seizures, can cause significant harm to neuronal cells through DNA, protein, and lipid damage, ultimately leading to cell death.

To assess seizure-induced ROS expression, we performed 4HNE staining in the hippocampal region. The results demonstrated a significant expression of ROS following pilocarpine-induced seizures. However, when amlodipine was administered, it effectively reduced ROS expression in the CA1, CA3, Sub, and DG regions (Figure 2A,B).

These findings indicate that amlodipine administration can effectively decrease ROS levels following seizures, suggesting its potential role in mitigating neuronal damage caused by oxidative stress.

### 3.3. Amlodipine Administration Increased Neuron Survival after Seizure

To evaluate the potential neuroprotective effect of amlodipine in pilocarpine-induced status epilepticus (SE), we stained neuronal nuclei (NeuN) to identify and quantify neurons in various regions of the hippocampus, including the CA1, CA3, Sub, and DG regions. Our objective was to compare the number of NeuN-positive neuron cells in the amlodipine-treated group and in the vehicle-treated group, to assess neuronal survival. After one week, the analysis of stained brain samples revealed a significantly higher number of live neuron cells in the amlodipine-treated SE group, compared to the SE vehicle group. This observation suggests that administering amlodipine for one week in rats has a neuroprotective effect, mitigating neuronal death caused by pilocarpine-induced status epilepticus.

The seizure-amlodipine group had a higher number of NeuN-positive cells compared to the seizure-vehicle group. NeuN is a neuronal marker, and NeuN-positive cells are typically used to quantify neurons in various conditions. Despite this increase, the number of NeuN-positive cells in the seizure-amlodipine group was still lower than each sham group. This implies that, while amlodipine may have had a protective or restorative effect on neuron numbers post-seizure, it did not fully restore neuron numbers to the level seen in the healthy control (sham) groups (Figure 3A,B).

These findings indicate that amlodipine may play a crucial role in preserving neuronal integrity and enhancing neuronal survival during status epilepticus, highlighting its potential as a therapeutic agent for protecting against the harmful effects of seizures on neuronal cells.

### 3.4. Amlodipine Reduced Blood–Brain Barrier Breakdown after Seizure

To explore amlodipine’s potential in safeguarding the blood–brain barrier (BBB) against pilocarpine-triggered seizures, we performed immunohistochemical examinations on brain tissues to measure serum IgG leakage, a marker of BBB impairment during seizures. Brain sections were immunostained for rat IgG, to compare leakage levels between the amlodipine- and vehicle-treated groups under seizure conditions. Our analysis revealed notably reduced IgG leakage in the amlodipine group, indicative of a significant protective effect of the drug on the BBB integrity during seizures. These results suggest that amlodipine may be instrumental in maintaining BBB health, potentially shielding the brain from deleterious agents and seizure-associated neuronal harm (Figure 4A,B).

Upon re-examination of the staining intensity data, we observed a notable variance in leakage among individual samples within the amlodipine-treated pilocarpine group. This variance is now more accurately represented in the revised graph in Figure 4, which shows the range of staining intensities observed, highlighting the significant difference in leakage when compared to the sham controls.

### 3.5. Amlodipine Administration Reduced Astrocyte Over-Activation after Seizure

In order to investigate the activation of astrocytes in the CA1 region of the hippocampus following pilocarpine-induced seizures, we conducted staining using glial fibrillary acidic protein (GFAP) and complement component 3 (C3). Astrocytes become activated in response to seizure-induced brain damage, leading to the expression of C3. The presence of C3 triggers synaptic death through phagocytosis by microglia or macrophages that recognize the C3 receptor, contributing to neuronal death and other seizure-related side effects.

The results of our staining analysis revealed that the seizure-vehicle group exhibited three times higher astrocyte activation than the sham groups. Similarly, the seizure-amlodipine group showed twice the level of astrocyte activation of the sham groups. Notably, there was a significant difference observed between the seizure-vehicle and seizure-amlodipine groups, in terms of astrocyte activation. The seizure-vehicle group had a lower number of DAPI-positive cells than the sham operating group. DAPI is a fluorescent stain that binds strongly to DNA, and it is commonly used in microscopy to visualize cell nuclei. This observation could imply a reduced number of cells in the seizure-vehicle group compared to the sham group (Figure 5A,B).

Furthermore, the intensity of C3 staining in the CA1 region was higher in the seizure-vehicle group compared to the other groups, while the seizure-amlodipine group exhibited similar intensity to the sham groups. This suggests that treating seizures with amlodipine significantly reduces the excessive activation of astrocytes and subsequent release of C3, which is associated with synaptic death in neurons [42].

Based on these findings, it can be concluded that amlodipine has a protective effect in mitigating astrocyte-mediated damage and neuronal death in the context of seizures. By reducing astrocyte activation and the release of C3, amlodipine may help preserve synaptic integrity and neuronal function, offering potential neuroprotective benefits during seizures.

### 3.6. Amlodipine Administration Reduced Microglia Over-Activation after Seizure

To evaluate the activation of microglia, we conducted staining using ionized calcium-binding adaptor molecule-1 (Iba1) and cluster of differentiation 68 (CD68), which specifically identify microglia and inflammatory microglia, respectively [43,44]. Microglia are the primary immune cells in the brain and play a crucial role in immune and inflammatory responses. However, the excessive activation of microglia after seizures can result in additional harm to neurons. In our study, we measured the intensity of Iba1 staining to assess microglia activation. The results demonstrated that the seizure-vehicle group exhibited approximately three times higher Iba1 intensity than the seizure-amlodipine group. Additionally, we measured the intensity of CD68 staining, to evaluate the presence of inflammatory microglia. The sham groups displayed similar intensities of CD68 staining, while the seizure-vehicle group showed around twice the intensity in comparison with the seizure-amlodipine group (Figure 6A,B).

These findings suggest that the treatment of seizures with amlodipine successfully inhibits microglia activation subsequent to the seizure episode. By reducing microglia activation, amlodipine may offer protective effects against secondary damage to neuronal cells following seizures.

### 3.7. Amlodipine Administration Reduced Microtubule Disruption after Seizure

To assess microtubule damage subsequent to seizures, we conducted staining for microtubule-associated protein 2 (MAP2) on brain samples, specifically focusing on the CA1 region of the hippocampus. Microtubules play a critical role in maintaining the cytoskeletal structure, and their disruption during seizures can lead to the increased vulnerability of brain tissue. Our results revealed that the sham groups exhibited comparable levels of MAP2 staining intensity, indicating intact microtubule structures. Conversely, the seizure groups displayed significantly lower positive intensities of MAP2, signifying that microtubule damage was a consequence of the seizures. However, in the seizure-amlodipine group, the intensity of MAP2 staining was twice that of the seizure-vehicle group. This observation suggests that amlodipine treatment was effective in preventing or mitigating microtubule damage caused by seizures (Figure 7A,B).

### 3.8. Amlodipine Administration Improved Spatial Cognitive, Memory, and Cognitive Function Recovery after Seizure

To assess the recovery of spatial cognitive ability and cognitive function following seizures, we employed the Barnes maze and adhesive removal behavior tests. These tests were conducted for five days, one week after inducing seizures or during a control period. Additionally, we stained brain samples using NeuN to evaluate overall neuron survival and the neuroprotective effects of amlodipine.

Our results showed that the sham groups displayed similar patterns in both the Barnes maze and adhesive removal tests. However, in the Barnes maze, the seizure groups did not exhibit any significant differences on days 1 and 2. From day 3 onwards, the seizure-amlodipine group not only displayed better performance in reaching the opening hole but also showed significant improvements compared to the seizure-vehicle group (Figure 8A). In the adhesive removal test, the seizure-amlodipine group exhibited significantly faster performance than the seizure-vehicle group on days 4 and 5. However, there were no significant differences between the seizure-vehicle group and the seizure-amlodipine group on days 1, 2, and 3 (Figure 8B).

We also conducted NeuN staining and quantified the number of NeuN-positive cells in the CA1 and CA3 regions of the hippocampus in brain samples. We found significant differences between the seizure-amlodipine group and the seizure-vehicle group, indicating enhanced neuron survival in the amlodipine-treated group. There were no significant differences observed in the sham groups (Figure 8C,D).

Overall, our findings suggest that amlodipine treatment after seizures improves the recovery of spatial cognitive ability and cognitive function. Furthermore, amlodipine appears to have a protective effect against microtubule damage induced by seizures. By preserving the integrity of microtubules, amlodipine may contribute to the stability and functionality of brain tissue, ultimately enhancing the recovery of spatial and functional cognitive abilities after seizures.

## 4. Discussion

The present study aimed to investigate the effects of amlodipine, which is known to block L-type voltage-gated calcium channels (LTCCs), on various aspects of neuronal damage and dysfunction caused by pilocarpine-induced seizures [45,46,47]. The results of this study revealed several positive effects of amlodipine treatment. Firstly, it was found that amlodipine treatment led to a reduction in the overexpression of Cav1.2 channels and zinc accumulation induced by seizures. These channels are involved in the entry of calcium (Ca^2+^) and zinc (Zn^2+^) ions into neurons during seizures, ultimately leading to neuronal damage [46,48]. The decreased expression of Cav1.2 channels in the amlodipine-treated group suggests that amlodipine inhibits the activation of these channels, potentially reducing the entry of harmful ions into neurons.

In addition, the study examined the effect of amlodipine treatment on reactive oxidative stress (ROS), which plays a significant role in neuronal damage during seizures [49,50,51,52,53,54,55]. It was observed that amlodipine treatment effectively reduced ROS levels in the hippocampal regions compared to the vehicle-treated group after seizures. By reducing ROS levels, amlodipine may mitigate oxidative damage to essential components in neuronal cells, such as DNA, proteins, and lipids, therefore protecting neuronal cells from harm.

Pilocarpine-induced seizures are a well-established experimental model for studying epilepsy, particularly temporal lobe epilepsy. When administered to animals, pilocarpine can induce seizures that closely resemble human temporal lobe epilepsy. These seizures often lead to neuronal death in the hippocampus, a critical brain region involved in memory and learning [5,6,34,56,57]. However, in the present study, we found that amlodipine treatment increased the survival of neurons following seizures. Staining for neuronal nuclei (NeuN) revealed a higher number of live neuron cells in the amlodipine-treated group than in the vehicle-treated group. This finding suggests that amlodipine treatment contributes to the preservation of neuronal viability during pilocarpine-induced status epilepticus.

Pilocarpine-induced seizures have been studied for their effects on blood–brain barrier (BBB) disruption, which is a critical aspect in understanding epilepsy and its complications. Research indicates that BBB breakdown can both induce epileptic seizures and result from them. This disruption is dynamic and time-dependent, and it is particularly noticeable in the acute phase of epilepsy induced by pilocarpine. Thus, we investigated whether amlodipine could prevent BBB disruption after a seizure [5,58,59,60,61,62]. In the present study, we found that amlodipine showed a protective effect on the blood–brain barrier (BBB) during seizures. The study examined BBB integrity using IgG leakage as an indicator, and the results showed that amlodipine treatment significantly reduced the intensity of IgG leakage, compared to the vehicle-treated group, after seizures [63,64,65,66]. This suggests that amlodipine protects against BBB disruption, preventing the passage of blood components into the brain tissue.

Research has shown that pilocarpine-induced seizures in rats led to the rapid activation of astrocytes and microglia in the brain. Astrocytes and microglia are types of glial cells in the central nervous system that play crucial roles in brain health and disease. In the context of epilepsy, such as in the pilocarpine-induced model, these cells become activated as part of the brain’s response to seizures. One study reported that, following pilocarpine-induced seizures, there was a significant increase in the labeling of astrocytes and microglial cells in the hippocampal region, indicating their activation. This suggests that glial cells respond quickly to seizure activity and may be involved in the pathophysiological changes that occur in the brain during epilepsy [67,68,69,70,71]. In the present study, we found that amlodipine treatment reduced the over-activation of astrocytes, as evidenced by the decreased expression of glial fibrillary acidic protein (GFAP) and complement component 3 (C3) staining. Additionally, the administration of amlodipine demonstrated the ability to prevent microglia activation in response to seizures. The excessive activation of astrocytes and microglia can cause secondary damage to neurons, so by inhibiting their over-activation, amlodipine may play a protective role in preventing such detrimental effects and preserving the integrity of neuronal cells.

Several studies have shown that pilocarpine-induced epilepsy causes microtubule-associated protein 2 (MAP2) disruption in the hippocampus [72,73]. Therefore, the present study examined the impact of amlodipine on microtubule disruption following seizures. It was observed that amlodipine treatment mitigated the damage caused to microtubules during seizures. Microtubules are essential for maintaining the structural integrity and stability of brain tissue, so preserving their integrity may contribute to the overall stability and function of neurons, offering protection against neuronal damage [72,74,75].

Calcium channels, particularly LTCCs, play a crucial role in various cognitive processes including memory formation, learning, and synaptic plasticity [76,77,78]. The influx of calcium through these channels is a key event in the signaling pathways that underlie these cognitive functions. Aberrant calcium signaling, often observed in neurodegenerative diseases, is linked to impaired cognitive abilities. Our study’s focus on amlodipine’s action on LTCCs, therefore, has significant implications for understanding and potentially treating cognitive dysfunctions associated with abnormal calcium channel activity. Research has shown that pilocarpine-induced status epilepticus in rats leads to significant cognitive impairment. This impairment includes deficits in memory and spatial cognition [6,62]. Thus, the present study evaluated the effect of amlodipine treatment on cognitive ability following seizures. Behavioral tests indicated that amlodipine treatment improved the recovery of cognitive function and spatial cognition in rats. Staining for neuronal nuclei (+) cells in various regions of the hippocampus further supported the hypothesis that amlodipine treatment had a positive impact on neuronal protection and cognitive ability following seizures.

This study highlights the critical role of calcium channels in neuronal function, especially in relation to seizures and their impact on cognitive abilities. The dysregulated influx of calcium into neurons during seizures, a result of abnormal neuronal firing, leads to an increase in intracellular calcium [79,80,81,82]. This increase can cause neuronal damage or death (excitotoxicity), disrupting neural circuits essential for cognitive functions such as memory, attention, and learning [83,84]. Our findings suggest that chronic alterations in calcium channel function and expression, as a result of prolonged or repeated seizures, could contribute to the long-term cognitive deficits often observed in chronic epilepsy [85]. The potential of calcium channel blockers in mitigating these seizure-induced cognitive impairments offers a promising therapeutic avenue [86].

A considerable body of research supports the effectiveness of calcium channel blockers in reducing the severity of various seizure types. This is evidenced in studies conducted by various researchers [87,88]. The dihydropyridine class of these blockers, in particular, has been noted for its anticonvulsant properties across multiple experimental settings [89,90,91]. A subset of these, the LTCC blockers, such as nicardipine, have been observed to provide neuroprotection under conditions of extended depolarization [92]. Nimodipine, another compound in this class, has demonstrated efficacy in mitigating seizure effects in various animal studies, including those focusing on pilocarpine-induced seizures in rats [47,90]. While the specific mechanisms of LTCC inhibitors are not completely understood, they are known to regulate calcium flow in and out of synaptosomes, affecting neurotransmitter dynamics [93]. Nimodipine’s neuroprotective effect may also play a role in moderating the excitotoxicity linked with an overproduction of free radicals during seizures induced by pilocarpine [94].

Our study provides insights into the action of amlodipine, a known L-type calcium channel (LTCC) inhibitor, across different physiological states. Particularly, we observed its varied efficacy in normal versus pathological conditions, such as those mimicking epilepsy. In pathological states characterized by elevated LTCC expression, amlodipine exhibited pronounced efficacy. This aligns with its role as an LTCC inhibitor, where heightened LTCC activity in disease states amplifies the drug’s impact. Contrary to its effect in pathological states, amlodipine showed no significant effects in normal physiological conditions. This finding is pivotal, underscoring the drug’s specificity and potential for targeted action in pathological states without influencing normal physiological processes. These observations emphasize the importance of context in pharmacological efficacy. Understanding how amlodipine functions differently under varying physiological conditions can guide its clinical use, ensuring effective and targeted therapy, especially in conditions with altered LTCC dynamics. In conclusion, our study highlights the context-dependent action of amlodipine, offering crucial insights for its application in therapeutic strategies that require precise modulation of calcium channels.

In summary, this study provides evidence supporting the neuroprotective effects of amlodipine in the context of pilocarpine-induced seizures. Amlodipine demonstrated the ability to reduce Cav1.2 channel overexpression and zinc accumulation, decrease reactive oxidative stress, increase neuron survival, reduce blood–brain barrier leakage, inhibit astrocyte over-activation, and protect against microtubule disruption. Furthermore, amlodipine improved the recovery of both spatial and functional cognitive ability following seizures. These findings collectively highlight the potential of amlodipine as a therapeutic intervention for mitigating neuronal damage and dysfunction associated with seizures.

While the present study provides insights into the neuroprotective effects of amlodipine in a pilocarpine-induced seizure model, it is important to consider: Firstly, the results are specific to the pilocarpine model of seizures, which may not fully replicate the complex pathophysiology of human epileptic conditions. Secondly, the findings in an animal model may not directly translate to human epilepsy treatment. Thirdly, the study may not have addressed the long-term effects and safety profile of amlodipine in the context of chronic epilepsy management. Fourthly, while the study indicates neuroprotective effects, the detailed mechanisms of how amlodipine affects neuronal functioning and epilepsy pathology might require further exploration.

## 5. Conclusions

The study highlights amlodipine’s potential in reducing brain damage from pilocarpine-induced seizures. These findings suggest its potential as a therapeutic agent.

## Figures and Tables

**Figure 1 antioxidants-13-00389-f001:**
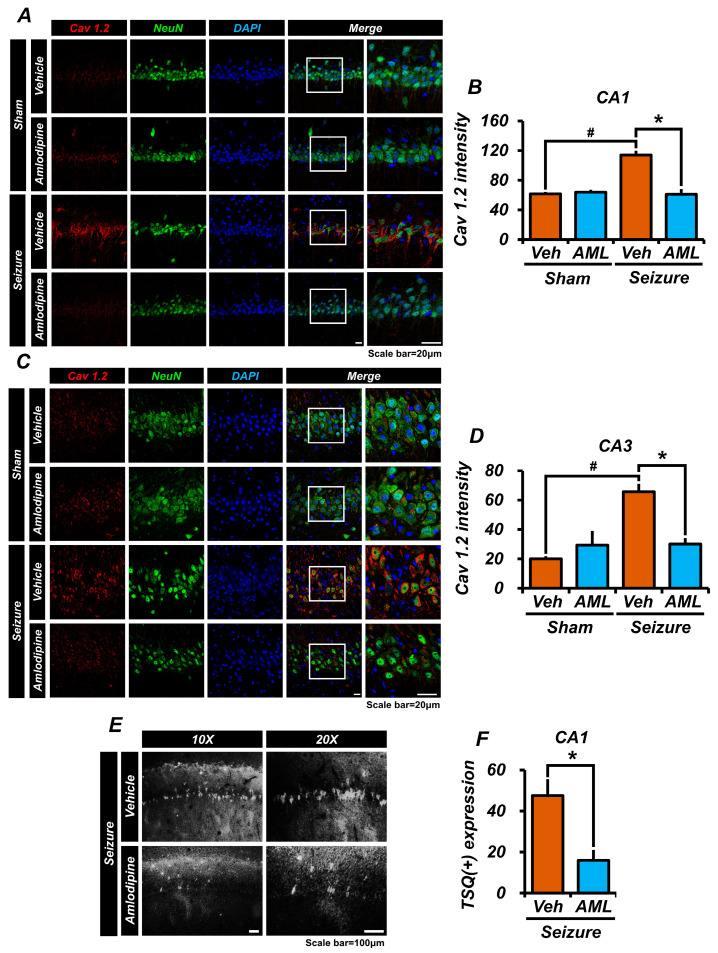
Amlodipine administration reduced Cav 1.2 activation and zinc accumulation in neurons following seizure. (**A**,**C**) Confocal micrographs of Cav1.2 (red), NeuN (green), and DAPI (blue) in the hippocampal CA1 and CA3 regions. Scale bar = 20 µm. (**B**,**D**) The bar graph shows the intensity of Cav1.2 in the hippocampal regions CA1 and CA3. Both graphs indicate that the intensity of Cav1.2 in the seizure-amlodipine group is similar to that of the sham groups and approximately half of that in the seizure-vehicle group, with significant differences. Additionally, there was no significant difference between the sham-vehicle group and the sham-amlodipine group. Animals were sacrificed on day 7. Cav 1.2 expression decreased in the seizure-amlodipine group, compared with seizure-vehicle group, by about 46% in theCA1 region (seizure-vehicle group, 113.9 ± 5.9; seizure-amlodipine group, 61.0 ± 7.2; sham-vehicle group, 61.6 ± 2.7; sham-amlodipine group, 64.0 ± 3.3), and by 54% in the CA3 region (seizure-vehicle group, 65.7 ± 5.5; seizure-amlodipine group, 30.0 ± 4.2; sham-vehicle group, 20.0 ± 1.7; sham-amlodipine group, 29.3 ± 9.7). The sample sizes were n = 3 animals for each sham group and n = 5 animals for each seizure group. (Bonferroni post hoc test after Kruskal–Wallis test, CA1: chi-squared = 10.212, df = 3, *p* = 0.017; CA3: chi-squared = 10.471, df = 3, *p* = 0.015.) (**E**) Difference in the number of TSQ (+) cells between the seizure-vehicle and seizure-amlodipine groups in the CA1 region one day after the seizure. (**F**) The seizure-amlodipine group shows a significant reduction in TSQ-positive neuron cells, with a decrease of about 66%, compared to the seizure-vehicle group. Thus, amlodipine treatment after a seizure reduces Cav1.2 over-activation and zinc accumulation, compared to vehicle treatment after a seizure. Animals were sacrificed on day 1. TSQ (+) neuron cells decreased in the seizure-amlodipine group, compared with seizure-vehicle group, by about 66% in the CA1 region (seizure-vehicle group, 47.5 ± 8.0; seizure-amlodipine group, 16.0 ± 5.0). Scale bar = 100 µm (the sample sizes were n = 5 animals for the seizure-amlodipine group, n = 6 animals for the seizure-vehicle group), (Mann–Whitney U test measurement results, whole brain region z = 2.373, *p* = 0.018). ^#^ indicates significant difference between sham vehicle and seizure vehicle groups, * Significantly different from the vehicle group, *p* < 0.05.

**Figure 2 antioxidants-13-00389-f002:**
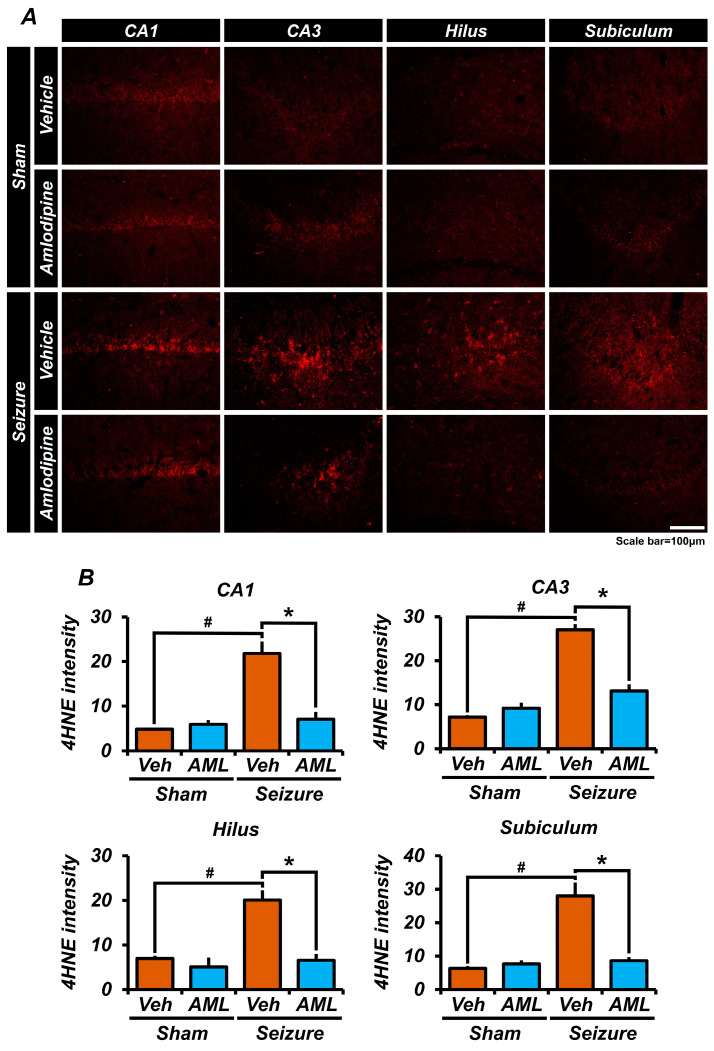
Administration of amlodipine decreased reactive oxidative stress (ROS) after seizure. ROS was detected using 4-hydroxy-2-nonenal (4HNE) (red) staining in the hippocampal CA1, CA3, dentate gyrus (DG), and subiculum (Sub) regions, 7 days after the seizure. (**A**) Micrographs showing 4HNE (red) in the hippocampal CA1, CA3, DG, and Sub regions. (**B**) In every region, there was no significant difference between the sham groups. There were differences in intensity between the seizure-vehicle and seizure-amlodipine groups in every region, including the CA3 region. In every region, the intensity was similar between the sham groups and the seizure-amlodipine-treated group, and there was also a significant difference between the sham-vehicle group and the seizure-vehicle group. Therefore, administering amlodipine after a seizure significantly reduced ROS, compared to administering the vehicle after a seizure. Animals were sacrificed on day 7. The expression of 4HNE decreased in the seizure-amlodipine group, compared with seizure-vehicle group, by about 68% in the CA1 region (seizure-vehicle group, 21.8 ± 3.1; seizure-amlodipine group, 7.0 ± 2.0; sham-vehicle group, 4.8 ± 0.3; sham-amlodipine group, 5.9 ± 1.0), by 52% in the CA3 region (seizure-vehicle group, 27.0 ± 2.4; seizure-amlodipine group, 13.1 ± 1.6; sham-vehicle group, 7.1 ± 0.5; sham-amlodipine group, 29.2 ± 1.3), by 69% in the hilus (seizure-vehicle group, 20.1 ± 3.4; seizure-amlodipine group, 6.6 ± 1.8; sham-vehicle group, 7.0 ± 0.6; sham-amlodipine group, 5.1 ± 2.1), and by 67% in the subiculum (seizure-vehicle group, 28.0 ± 4.0; seizure-amlodipine group, 8.6 ± 1.2; sham-vehicle group, 6.3 ± 0.7; sham-amlodipine group, 7.6 ± 1.1). Scale bar = 100 µm (the sample sizes were n = 5 animals for each sham group, n = 6 animals for the seizure-vehicle group, and n = 8 animals for the seizure-amlodipine group). (Bonferroni post hoc test after Kruskal–Wallis test, CA1: chi-squared = 11.085, df = 3, *p* = 0.011; CA3: chi-squared = 14.367, df = 3, *p* = 0.002; subiculum: chi-squared = 13.485, df = 3, *p* = 0.004; dentate gyrus: chi-squared = 12.686, df = 3, *p* = 0.005). ^#^ indicates significant difference between sham vehicle and seizure vehicle groups, * Significantly different from the vehicle group, *p* < 0.05.

**Figure 3 antioxidants-13-00389-f003:**
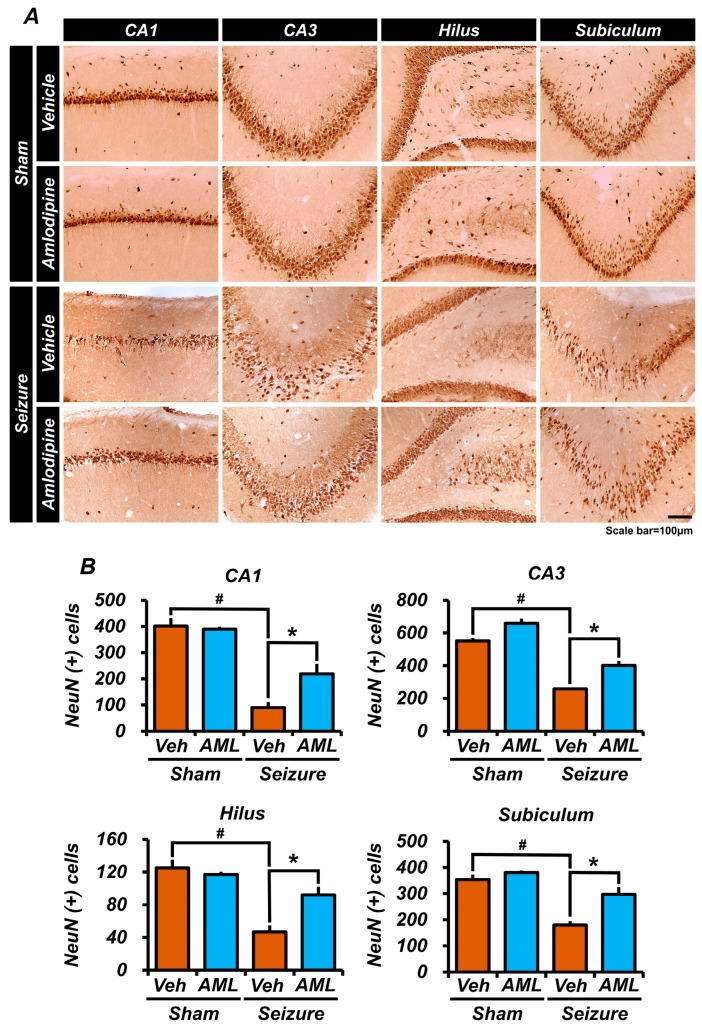
Amlodipine administration increased neuron survival after seizure. (**A**) Micrographs of NeuN (+) cells representing surviving cells in the hippocampal CA1, CA3, hilus, and subiculum (Sub) regions. The number of surviving neurons appears to be greater in the seizure-amlodipine group than in the seizure-vehicle group. (**B**) Cell counting in the micrographs of every region shows a significant difference between the sham-vehicle group and the seizure-vehicle group. In the CA1 and hilus regions, cell counting showed that the seizure-amlodipine group had approximately twice the number of NeuN (+) cells as the seizure-vehicle group. In the CA3 and Sub regions, the seizure-amlodipine group had more NeuN (+) cells, which was a noticeable difference from the seizure-vehicle group. Additionally, in every region, there is no significant difference in NeuN (+) cells between the sham-vehicle group and the sham-amlodipine group. Therefore, administering amlodipine after a seizure significantly increases neuron survival compared to treating with a vehicle after a seizure. Animals were sacrificed on day 7. The number of NeuN (+) cells was higher in the seizure-amlodipine group than the seizure-vehicle group by about 58% in the CA1 region (seizure-vehicle group, 89.5 ± 21.5; seizure-amlodipine group, 215.4 ± 32.3; sham-vehicle group, 401.6 ± 22.3; sham-amlodipine group, 389.5 ± 6.9), by 34.3% in the CA3 region (seizure-vehicle group, 257.9 ± 8.7; seizure-amlodipine group, 392.8 ± 24.7; sham-vehicle group, 551.1 ± 28.3; sham-amlodipine group, 659.0 ± 19.4), by 49% in the hilus region (seizure-vehicle group, 46.7 ± 8.0; seizure-amlodipine group, 92.1 ± 10.0; sham-vehicle group, 125.2 ± 11.0; sham-amlodipine group, 117.0 ± 2.8), and by 39% in the subiculum (seizure-vehicle group, 179.9 ± 14.1; seizure-amlodipine group, 297.0 ± 25.4; sham-vehicle group, 353.8 ± 14.9; sham-amlodipine group, 380.3 ± 6.6). Scale bar = 100 µm (the sample sizes were n = 5 animals for each sham group, n = 6 animals for the seizure-vehicle group, and n = 8 animals for the seizure-amlodipine group). (Bonferroni post hoc test after Kruskal–Wallis test, CA1: chi-squared = 17.649, df = 3, *p* = 0.001; CA3: chi-squared = 18.543, df = 3, *p* < 0.001; subiculum: chi-squared =15.535, df = 3, *p* = 0.001; dentate gyrus: chi-squared = 15.649, df = 3, *p* < 0.001.) ^#^ indicates significant difference between sham vehicle and seizure vehicle groups, * Significantly different from the vehicle group, *p* < 0.05.

**Figure 4 antioxidants-13-00389-f004:**
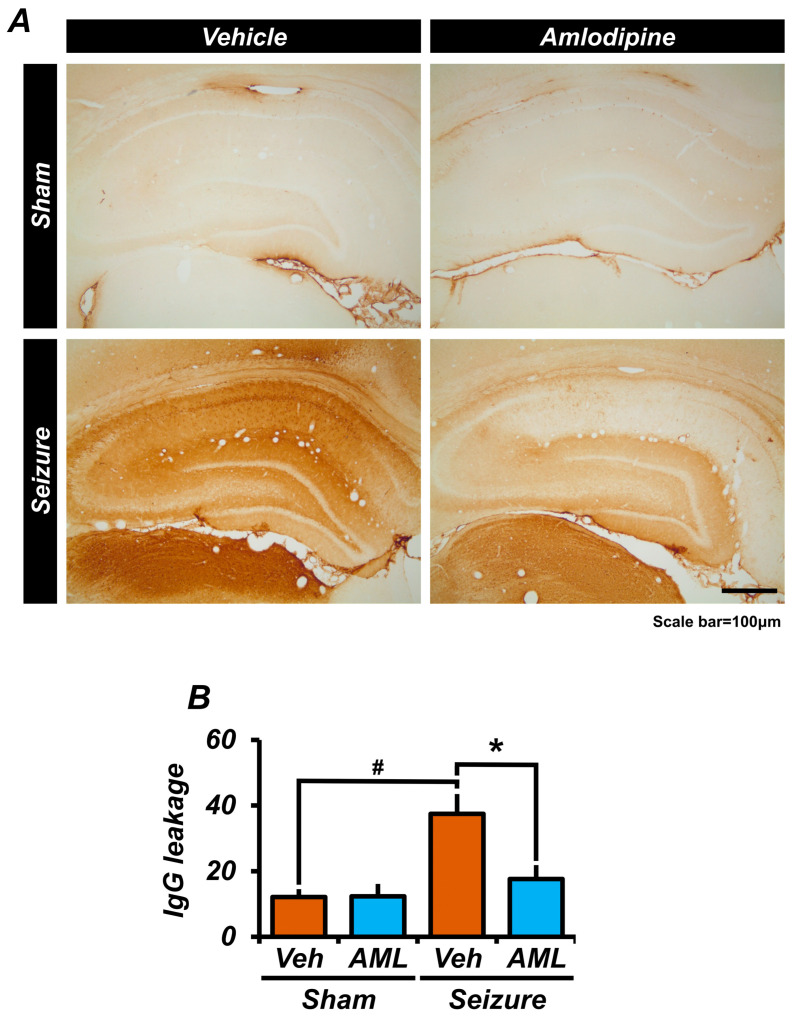
Amlodipine administration reduced blood–brain barrier (BBB) breakdown after seizure. (**A**,**B**) The micrographs show the amount of IgG leakage in a specific region of the whole brain. We found a significant difference between the sham-vehicle group and the seizure-vehicle group. There is also a twofold decrease in IgG leakage between the seizure-amlodipine group and the seizure-vehicle group. However, the IgG leakage in the seizure-amlodipine group is similar to that of the sham groups. Also, the sham-vehicle group shows no significant difference when compared with the sham-amlodipine group. This indicates that administering an amlodipine treatment after a seizure significantly reduces IgG leakage. The scale bar represents a distance of 100 µm. The data are presented as mean values with the standard error of the mean (SEM) indicated. The sample sizes were n = 3 animals for each sham group, n = 5 animals for the seizure-vehicle group, and six for the seizure-amlodipine group. The difference between the seizure-amlodipine group and the seizure-vehicle group is considered statistically significant, with a *p*-value less than 0.05. Animals were sacrificed on day 7. IgG leakage decreased in the seizure-amlodipine group, compared with the seizure-vehicle group, by about 53% in the hippocampal area (seizure-vehicle group, 37.4 ± 6.0; seizure-amlodipine group, 17.6 ± 4.3; sham-vehicle group, 12.1 ± 2.4; sham-amlodipine group, 12.3 ± 3.8). (Bonferroni post hoc test after Kruskal–Wallis test of IgG, CA1: chi-squared = 7.763, df = 3, *p* = 0.051.) ^#^ indicates significant difference between sham vehicle and seizure vehicle groups, * Significantly different from the vehicle group, *p* < 0.05.

**Figure 5 antioxidants-13-00389-f005:**
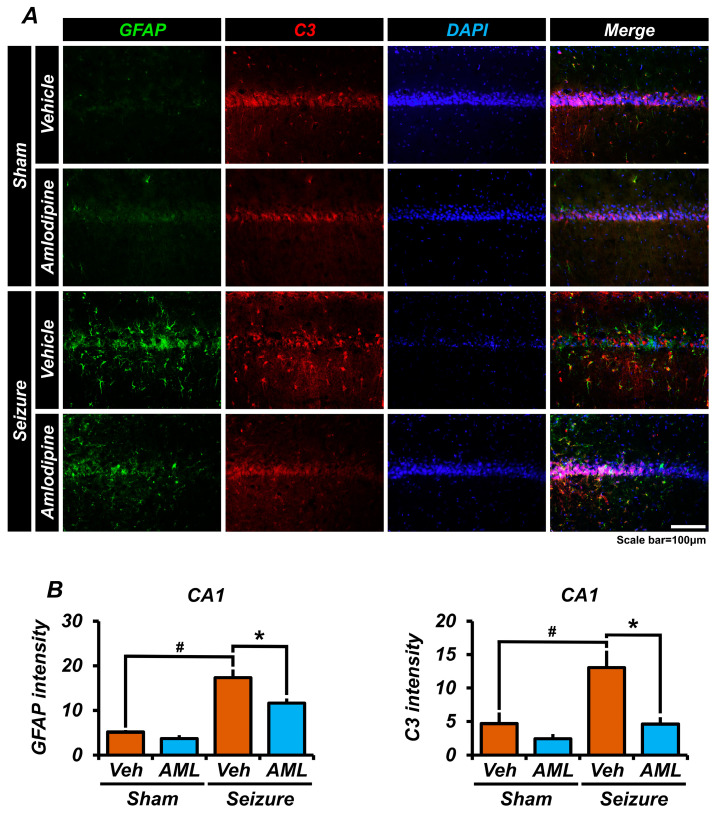
Amlodipine administration reduced astrocyte over-activation after seizure. (**A**) The micrographs show the staining of GFAP (green), C3 (red), and DAPI (blue) in the hippocampal CA1 region. (**B**) The bar graph illustrates the intensity of GFAP, which is significantly different in the sham-vehicle group compared with the seizure-vehicle group; there is also a significant difference between the seizure-amlodipine group and the seizure-vehicle group. The intensity of C3 in the seizure-amlodipine group is comparable to each sham group and is approximately three times lower than that of the seizure-vehicle group. There is also a significant difference between the sham-vehicle group and the seizure-vehicle group. Also, there is no significant difference in GFAP and C3 intensity among the various sham groups. This indicates that administering an amlodipine treatment after a seizure reduces the over-activation of astrocytes. The scale bar represents a distance of 100 µm. The data are presented as mean values, with the standard error of the mean (SEM) indicated. The sample sizes were five animals for each sham group, six animals for the seizure-vehicle group, and eight animals for the seizure-amlodipine group. The difference between the seizure-amlodipine group and the seizure-vehicle group is considered statistically significant, with a *p*-value less than 0.05. Animals were sacrificed on day 7. GFAP and C3 intensity decreased in the seizure-amlodipine group, compared with the seizure-vehicle group, by about 33% in the CA1 region for GFAP (seizure-vehicle group, 17.3 ± 1.8; seizure-amlodipine group, 11.6 ± 1.0; sham-vehicle group, 5.1 ± 0.4; sham-amlodipine group, 3.7 ± 0.8) and by 64% in the CA1 region for C3 (seizure-vehicle group, 17.3 ± 2.5; seizure-amlodipine group, 6.2 ± 1.4; sham-vehicle group, 5.5 ± 1.9; sham-amlodipine group, 12.9 ± 0.8). (Bonferroni post hoc test after Kruskal–Wallis test for GFAP, CA1: chi-squared = 12.894, df = 3, *p* = 0.005; Bonferroni post hoc test after Kruskal–Wallis test for C3, CA1: chi-squared = 12.246, df = 3, *p* = 0.007.) ^#^ indicates significant difference between sham vehicle and seizure vehicle groups, * Significantly different from the vehicle group, *p* < 0.05.

**Figure 6 antioxidants-13-00389-f006:**
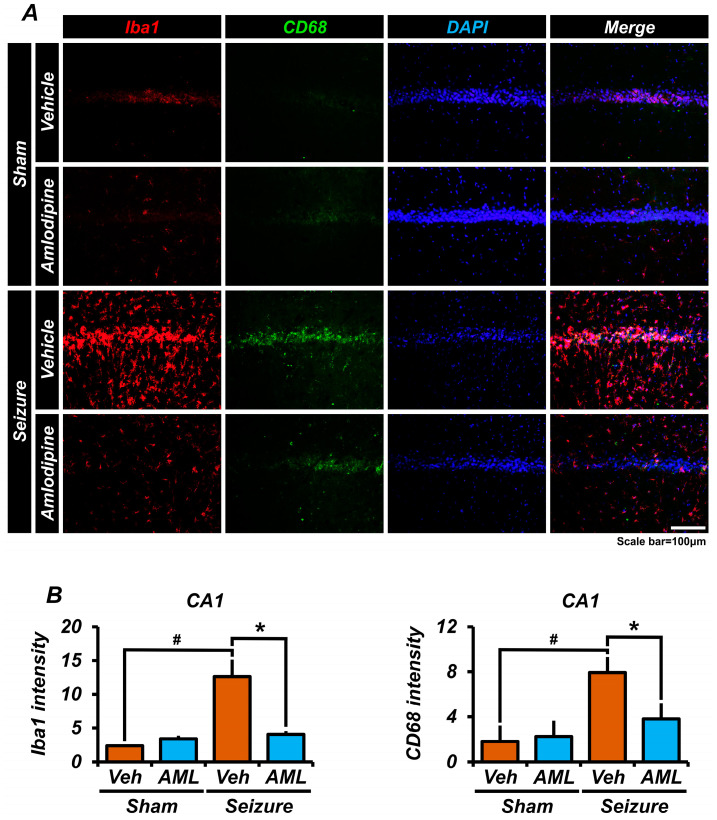
Amlodipine administration reduced microglia over-activation after seizure. (**A**) The micrographs depict the staining of Iba1 (red), CD68 (green), and DAPI (blue). There is a high presence of detected cells positive for both Iba1 and CD68 in the seizure-vehicle group. (**B**) The bar graph displays the intensity of Iba1 staining, which is found to be twice as high in the seizure-vehicle group compared to the other groups. The intensity of Iba1 in both the seizure-amlodipine group and each sham group is almost similar. Additionally, there is a significant difference in C3 intensity between the seizure-vehicle group and the seizure-amlodipine group, with the latter showing a slightly higher intensity compared to each sham group. Additionally, both graphs illustrate a significant difference between the sham-vehicle group and the seizure-vehicle group, while showing no significant difference among the sham groups. This indicates that administering amlodipine after a seizure reduces the over-activation of astrocytes, in comparison to the seizure-vehicle group. The scale bar represents a distance of 100 µm. The data are presented as mean values with the standard error of the mean (SEM) indicated. The sample sizes were five animals for each sham group, six animals for the seizure-vehicle group, and eight animals for the seizure-amlodipine group. The difference between the seizure-amlodipine group and the seizure-vehicle group is considered statistically significant, with a *p*-value less than 0.05. Animals were sacrificed on day 7. Iba1 and CD68 intensity decreased in the seizure-amlodipine group, compared with the seizure-vehicle group, by about 68% in the CA1 region for Iba1 (seizure-vehicle group, 12.6 ± 2.5; seizure-amlodipine group, 4.1 ± 0.5; sham-vehicle group, 2.4 ± 0.1; sham-amlodipine group, 3.4 ± 0.7) and by 52% in the CA1 region for CD68 (seizure-vehicle group, 7.9 ± 1.6; seizure-amlodipine group, 3.8 ± 0.7; sham-vehicle group, 1.8 ± 0.3; sham-amlodipine group, 2.3 ± 0.5). (Bonferroni post hoc test after Kruskal–Wallis test for Iba1, CA1: chi-squared = 16.884, df = 3, *p* = 0.001; Bonferroni post hoc test after Kruskal–Wallis test for CD68, CA1: chi-squared = 12.379, df = 3, *p* = 0.006.) ^#^ indicates significant difference between sham vehicle and seizure vehicle groups, * Significantly different from the vehicle group, *p* < 0.05.

**Figure 7 antioxidants-13-00389-f007:**
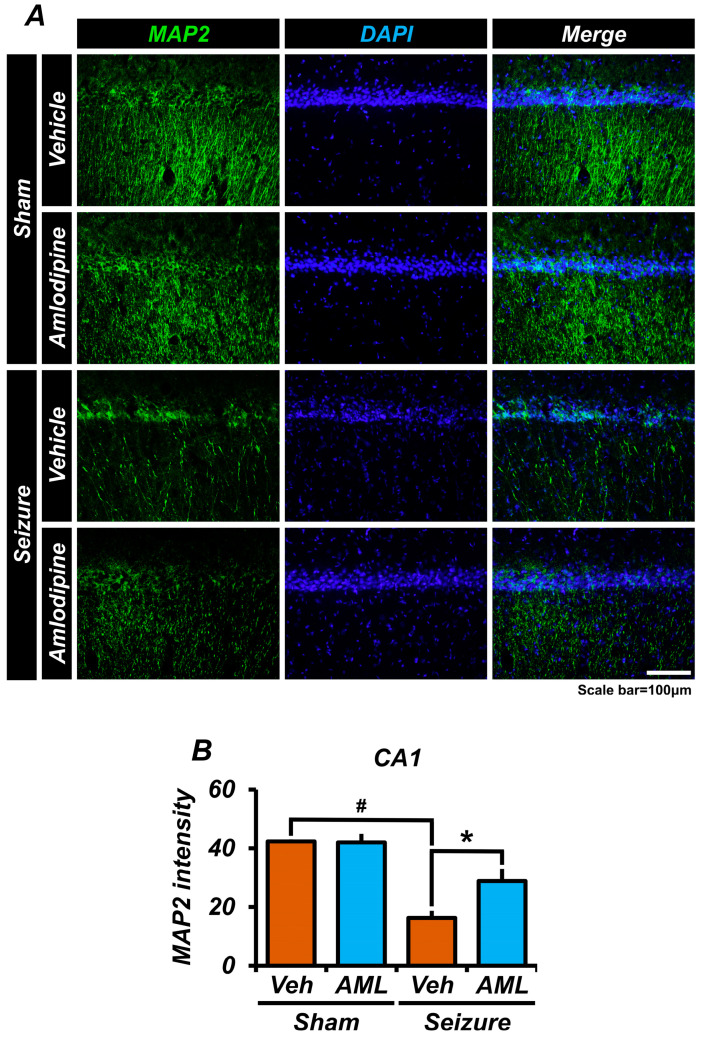
Amlodipine administration reduced microtubule disruption after seizure. (**A**) The micrographs illustrate the staining of MAP2 (green) and DAPI (blue). The intensity of MAP2 staining in the seizure groups is lower than in the sham groups. (**B**) The bar graph demonstrates that each sham group exhibits a similar intensity of MAP2 staining. There is a significant difference between the sham-vehicle group and the seizure-vehicle group, and the seizure-amlodipine group has a lower intensity than the sham groups but a higher intensity than the seizure-vehicle group, with a significant difference observed. Additionally, there was no significant difference between the sham-vehicle group and the sham-amlodipine group. These results indicate that administering amlodipine after a seizure provides protection to microtubules. The scale bar represents a distance of 100 µm. The data are presented as mean values, with the standard error of the mean (SEM) indicated. The sample sizes were five animals for each sham group, six animals for the seizure-vehicle group, and eight animals for the seizure-amlodipine group. The difference between the seizure-amlodipine group and the seizure-vehicle group is considered statistically significant, with a *p*-value less than 0.05. Animals were sacrificed on day 7. Microtubule intensity increased in the seizure-amlodipine group, compared with the seizure-vehicle group, by about 43% in the CA1 area (seizure-vehicle group, 16.3 ± 2.4; seizure-amlodipine group, 28.9 ± 4.1; sham-vehicle group, 42.4 ± 0.7; sham-amlodipine group, 42.0 ± 2.9). (Bonferroni post hoc test after Kruskal–Wallis test, CA1: chi-squared =10.452, df = 3, *p* = 0.015.) ^#^ indicates significant difference between sham vehicle and seizure vehicle groups, * Significantly different from the vehicle group, *p* < 0.05.

**Figure 8 antioxidants-13-00389-f008:**
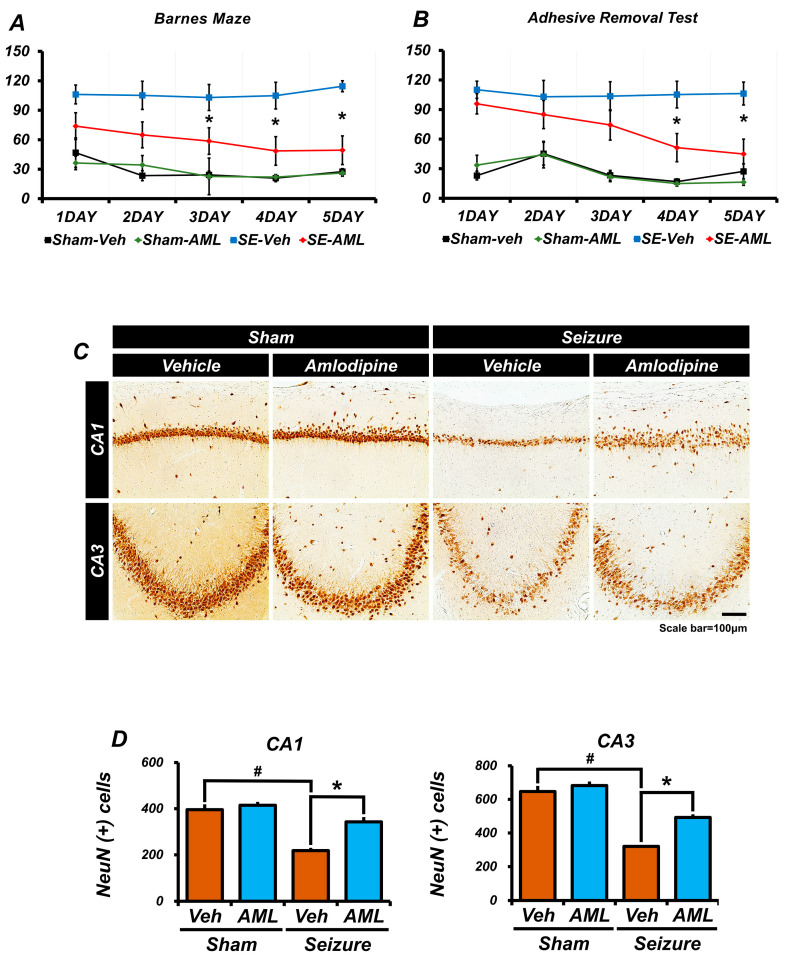
Amlodipine administration improved spatial cognitive, memory, and cognitive function recovery after seizure. One week following the seizures for the seizure groups, and after a recovery period for the sham groups, we conducted several tests, and the results were as follows: (**A**) Barnes maze test: This test assessed spatial cognitive ability recovery. The sham groups outperformed the seizure groups. Notably, the seizure-amlodipine group performed better than the seizure-vehicle group, with the most significant differences observed between the third and fifth day. Additionally, there was no significant difference between the sham-vehicle group and the sham-amlodipine group on any day. Statistical analyses (a repeated measures test, followed by an ANOVA) revealed significant differences, especially in group interaction effects. (Barnes maze: time x group: F = 3.268, *p* < 0.021.) (**B**) Adhesive removal test: This measured cognitive function recovery, and the seizure-amlodipine group demonstrated progressive improvement from day one to five. In contrast, the seizure-vehicle group exhibited negligible progression, with stark differences apparent on the fourth and fifth days. Additionally, there was no significant difference between the sham groups on any day. Statistical analyses (a repeated measures test, followed by an ANOVA) revealed significant differences, especially in group interaction effects (adhesive removal: time * group: F = 12.005, *p* < 0.006). (**C**) We analyzed micrographs for neuronal nuclei (NeuN)-positive cells in the hippocampal CA1 and CA3 areas. (**D**) Bar graphs showed the sham groups had similar NeuN (+) cell counts in the hippocampal regions. In contrast, there was a noticeable discrepancy in the seizure groups, with a significant difference compared to the sham-vehicle group, as the amlodipine treatment seemed to bolster recovery in spatial cognition and general cognitive functions. Also, there were no differences in the CA1 and CA3 regions between the various sham groups. Each sham group consisted of five animals, while the seizure groups had six. Statistical significance was achieved between the seizure-amlodipine and seizure-vehicle groups, with a *p*-value below 0.05. Animals were sacrificed on day 12. The number of NeuN (+) cells was higher in the seizure-amlodipine group, compared with the seizure-vehicle group, by about 36% in the CA1 region (seizure-vehicle group, 218.2 ± 12.1; seizure-amlodipine group, 342.7 ± 21.1; sham-vehicle group, 395.8 ± 23.1; sham-amlodipine group, 414.9 ± 14.4) and by 35% in the CA3 region (seizure-vehicle group, 320.4 ± 5.3; seizure-amlodipine group, 492.5 ± 18.8; sham-vehicle group, 646.6 ± 33.5; sham-amlodipine group, 682.3 ± 24.2). (Bonferroni post hoc test after Kruskal–Wallis test: CA1: *p* = 0.006; CA3: *p* = 0.004.) This underscores the neuroprotective potential of amlodipine post-seizure. ^#^ indicates significant difference between sham vehicle and seizure vehicle groups, * Significantly different from the vehicle group, *p* < 0.05.

## Data Availability

All data generated or analyzed during this study are included in this published article.

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
